# Bridges-Round 2: A study protocol to examine the longitudinal HIV risk prevention and care continuum outcomes among orphaned youth transitioning to young adulthood

**Proscovia Nabunya[1,2]\***, **Ozge Sensoy Bahar[1,2]**, **Torsten B. Neilands[3]**, **Noeline Nakasujja[4]**, **Phionah Namatovu[2]**, **Flavia Namuwonge[1,2]**, **Abel Mwebembezi[5]**, **Fred M. Ssewamala[1,2]**

1 International Center for Child Health and Development (ICHAD), Brown School, Washington University in St. Louis, St. Louis, MO, United States of America, 2 International Center for Child Health and Development (ICHAD), Uganda Field Office, Masaka, Uganda, 3 Department of Medicine, Division of Prevention Science, Center for AIDS Prevention Studies, University of California, San Francisco, CA, United States of America, 4 Department of Psychiatry, School of Medicine, College of Health Sciences, Makerere University, Kampala, Uganda, 5 Reach the Youth Uganda, Kampala, Uganda

\* nabunyap@wustl.edu

**Data Availability Statement:** No datasets were generated or analysed during the current study. All

## Abstract

### Background

Youth orphaned by HIV in sub–Saharan Africa experience immense hardships including social disadvantage, adverse childhood events and limited economic prospects. These adversities disrupt the normative developmental milestones and can gravely compromise their health and emotional wellbeing. The *Bridges to the Future* study (2012–2018) prospectively followed 1,383 adolescents, between 10–16 years, to evaluate the efficacy and cost-effectiveness of a family-based economic empowerment intervention comprising of child development accounts, financial literacy training, family income generating activities and peer mentorship. Study findings show efficacy of this contextually-driven intervention significantly improving mental health, school retention and performance and sexual health. However, critical questions, such as those related to the longitudinal impact of economic empowerment on HIV prevention and engagement in care remain. This paper presents a protocol for the follow-up phase titled, *Bridges Round 2*.

### Methods

The *Original Bridges* study participants will be tracked for an additional four years (2022–2026) to examine the longitudinal developmental and behavioral health outcomes and potential mechanisms of the effect of protective health behaviors of the *Bridges* cohort. The study will include a new qualitative component to examine participants' experiences with the intervention, the use of biomedical data to provide the most precise results of the highly relevant, but currently unknown sexual health outcomes among study participants, as well as a cost-benefit analysis to inform policy and scale-up.

relevant data from this study will be made available upon study completion.

**Funding:** The study outlined in this protocol is funded by the National Institute of Mental Health (NIMH) under award # R01MH128232 (MPIs: Fred M. Ssewamala, PhD; Proscovia Nabunya, PhD, Ozge Sensoy Bahar, PhD). NIMH has no role in the study design, data collection, analysis, interpretation of findings and manuscript preparation. The content is solely the responsibility of the authors and does not necessarily represent the official views of NIMH or the National Institutes of Health.

**Competing interests:** The authors have declared that no competing interests exist.

## Discussion

Study findings may contribute to the scientific knowledge for low-resource communities on the potential value of providing modest economic resources to vulnerable boys and girls during childhood and early adolescence and how these resources may offer long-term protection against known HIV risks, poor mental health functioning and improve treatment among the HIV treatment care continuum.

## Introduction

Globally, adolescents and young adults between 14–29 years represent 30% of HIV incidence cases among persons of reproductive age, with the vast majority of cases (~75%) occurring in sub–Saharan Africa (SSA) [1]. HIV incidence rises rapidly in SSA, as adolescents and young adults leave school (often prematurely) and migrate for work and marriage [2–4]. In particular, older adolescents are more likely to experience increased experimentation with health-compromising behaviors that increase their vulnerability to HIV and sexually transmitted infections (STIs), including sexual risk behaviors and alcohol/ drug misuse [5–7]. Young people orphaned by AIDS (YPoAIDS), 80% (~10 million) of which live in SSA represent a particularly vulnerable and unique population [8]. Between 2012–2018, our *Bridges to the Future* study team prospectively followed 1,383 adolescents (10–16 years at study enrollment, who lost one or both parents to AIDS), across 48 primary schools in Uganda, to evaluate the efficacy and cost-effectiveness of a family-based economic empowerment intervention (EEI) comprising of child development accounts, financial literacy training, family income generating activities and peer mentorship. Our findings show short-term success with reduction in multidimension poverty incidence ($b$–0.107, SE = 0.034, p<0.001) [9], improving self-related health (B = 0.25, 95% CI = 0.06–0.43) and emotional well-being, including lowered depression (B = -0.28, 95% CI = -0.43–0.125), hopelessness (B = 0.05, 95% CI = –0.11, 0.21), improved self-concept (B = 0.26, 95% CI 0.09–0.44), and adherence self-efficacy (B = 0.26, 95% CI = 0.09–0.43); reduced self-reported sexual risk-taking behaviors (B = 0.05, 95% CI = –0.11, 0.21) [10,11]; and positive educational outcomes among intervention participants, including better academic performance ($b = -3.78$, 95% CI = –4.92, –2.64, $p \leq 0.001$) and higher odds of transitioning to post primary education (OR = 1.66, 95% CI = 1.28, 2.18, $p \leq 0.001$) [9–12]. However, critical questions related to the longitudinal impact of economic empowerment on HIV prevention and engagement in care among YPoAIDS remain.

This *Bridges-Round 2* (hereafter *Bridges-R2*) study proposes to examine the longitudinal impacts of the *Bridges* intervention on HIV risk prevention and care continuum outcomes among orphaned youth transitioning to young adulthood, a period characterized by unique vulnerabilities including increased experimentation, lower social control, and treatment non-adherence for those living with HIV [13,14]. Guided by the life course perspective [15], *Bridges-R2* will build on the *Original Bridges to the Future* study, to examine the longitudinal developmental and health outcomes [e.g., sexual risk-taking behaviors, HIV antiretroviral therapy adherence, pre-exposure prophylaxis (PrEP) use (for those who are HIV negative), optimizing health outcomes along the HIV care continuum, and potential mechanisms of effect of protective health behaviors] of the *Bridges* cohort. The study is guided by the following specific aims:

**Aim 1.** To examine the long-term impact of *Bridges* on: a) HIV prevalence (measured via participant's HIV status) (primary outcome); and b) Explore in secondary analyses the long-term impact of *Bridges* on key developmental and behavioral outcomes (e.g., mental health, alcohol and drug misuse).

**Aim 2**. To elucidate the long-term effects of *Bridges* on potential mechanisms of change, including: a) economic stability, viral suppression (for adolescents living with HIV); PrEP use (for HIV negative adolescents), medical male circumcision (for boys); and b) young adult transitions.

**Aim 3:** To qualitatively investigate participants' experiences with *Bridges* that may have influenced engagement with the program, sexual risk-taking decisions, financial behaviors; experiences with developmental transitions; and perceptions on program sustainability.

**Aim 4:** To assess the long-term costs and benefits of *Bridges* using formal economic evaluation.

## Background

Adolescents in SSA suffer more from poverty and high HIV prevalence rates than in all other regions in the world. Despite advances in prevention efforts, including treatment as prevention, HIV testing, and PrEP [16,17], HIV incidence for youth remains high [18] and HIV prevention efforts often remain low [16]. Adolescents living with HIV/AIDS face a) more critical developmental, psychosocial and economic adversities, b) less social support and opportunities for education, and c) compounded risk for mental health, addiction, and HIV-risk struggles than the general population [19–21]. Moreover, an understudied hardship of HIV/AIDS in SSA is the high prevalence of young people, including adolescents, who have lost one or both parents to this virus. Worldwide, there are ~20 million YPoAIDS and nearly 12 million are in SSA [22]. Growing up orphaned poses tremendous challenges, including higher rates of HIV risk behavior, higher odds of acquiring HIV infection and low likelihood to test for HIV than non-orphans [23–25].

Uganda, a fragile and low-resource country in SSA, is highly afflicted with poverty and HIV/AIDS. With an estimated 47 million people, 34.5% of the population live on less than $1.90 a day [26]. Uganda is one of the top 10 countries with the largest number of poor people in Africa [26]. The poverty rate is higher in rural areas–such as the proposed study region, estimated at 33.8% compared to 19.8% in urban areas [27]. Among children, an estimated 56% of Uganda's children below the age of 18 experience multidimensional deprivation and a low standard of living [28]. In addition, approximately 1.4 million Ugandans are currently living with HIV, ~53,000 new HIV infections occurred in 2019, and nearly 21,000 people died in 2017 because of HIV/AIDS [29]. Moreover, Uganda is highly impacted by orphanhood, with ~1.2 million children orphaned due to AIDS [30]. Young people in Uganda, especially those affected by AIDS, often live in poverty and show high rates of depression [31–33], anxiety, learning challenges [34,35], and sexual risk-taking [36,37]. They often experience stigma related to HIV orphanhood [38], low self-esteem and hopelessness about their future, which can negatively influence decisions about substance use and sexual risk-taking, further increasing HIV vulnerability [39]. Moreover, young people in poverty-impacted families have lower levels of secondary education, higher rates of low-wage work, and more young parenting [40,41], which result in negative outcomes such as unstable housing, substance use, mental health problems, and sexual risk-taking leading to HIV/AIDS [40–42], all of which compromise successful social transitions.

### Emerging adulthood as a precarious developmental period

Emerging adulthood typically defined as ages 18–25 [13], is a precarious developmental period requiring effective early-in-life prevention interventions to promote positive behavioral and health outcomes. Emerging adulthood is characterized by identity formation as well as a feeling of existing "in between" adolescence and adult-stages [14]. It is one of the most challenging

transition periods [14]. The ordering, timing, and tempo of these transitions can have health consequences, including HIV risk behaviors, and suboptimal adherence to long-term prescribed medications for youth living with HIV (YLHIV) [40]. Moreover, this age group is also at a higher risk of poor mental health functioning [13,43–45]. Successfully accomplishing developmental tasks during emerging adulthood not only influences immediate functioning, but also lays the foundation for optimal functioning later in life [46,47]. Orphaned youth are particularly vulnerable during this transition period. They are subject to permanent separation from biological parents, multiple relocations, stigma and discrimination associated with orphanhood [29,31,42,48,49]. Moreover, lack of family support may make developmental transitions more difficult [50–52]. Yet, little is known about how YPoAIDS and YLHIV navigate transition into young adulthood. The *Bridges* study of YPoAIDS (recruited at ages 10–16 years) from poor communities in Uganda provides a strategic opportunity to examine this transition period, identify key predictors of positive transitions among YPoAIDS, including YLHIV, and isolate the effect of financial instability on their transition milestones and behavioral health.

## Theoretical frameworks guiding the study

**Developmental and emerging adulthood.** Orphaned youth are subject to a host of risks including, permanent separation from biological parents, often multiple relocations, stigma and discrimination [29,31,42,48,49]. These experiences may interfere with the normal development and disrupt the healthy transition through developmental stages, resulting in disengagement from opportunities, school failure, risk-taking behaviors, poor emotional well-being, and poverty [50,51]. Moreover, lack of family support may make developmental transitions more difficult [50,51]. Thus, interventions aimed at supporting youth living with HIV should focus on increasing self-efficacy and enhanced control over one's life [50,51].

Within the developmental theoretical framework is the emerging adulthood transitional period, with its associated unique characteristics and vulnerabilities. Emerging adulthood is characterized with instability, constant and quick changes as well as heterogeneity in the pace and order of transitional milestones among young people. It is a period that involves five major role transitions: leaving home, completing school, entering the workforce, forming a romantic partnership, and transitioning to parenthood [15,46]. It is a period when youth initiate adult roles and responsibilities [53–55] and establish patterns of positive and risky health behaviors that carry through to adulthood [56–58]. Specific to SSA, a region heavily impacted by HIV and poverty, emerging adulthood is particularly important because it is when individuals become increasingly sexually active, and thus are at an elevated risk of contracting HIV [43]. YPoAIDS are particularly vulnerable during this life stage because they are often subject to multiple relocations, and stigma and discrimination related to HIV/AIDS orphanhood [29,31,42,48,49]. This study offers a time-sensitive and unique opportunity to examine how small direct investments through an asset-oriented EEI package received during adolescence may have long-term effects as youth transition into emerging adulthood.

**Asset theory.** Asset theory posits that assets can lead to wide scale benefits, including expectations for more resources in the future, feelings of safety and security [59], and future planning [60,61]. Asset-building is increasingly viewed as a critical factor for reducing poverty, positively impacting attitudes and behaviors, and improving psychosocial functioning and stability [62–65]. Asset theory is consistent with several behavioral and psychosocial theories that have guided research on sexual risk-taking and health and mental health functioning, including the Bandura Social Cognitive Theory [66] and the Theory of Reasoned Action [67,68]. Asset-theory contributes to our understanding of how attitudes and beliefs evolve, thus influencing intentions and behaviors [59].

| Intervention | Description |
|---|---|
| *Asset building; Planning* | Both treatment arms received twelve 1–2-hour sessions, covering topics on: a) asset-building, saving, and investing; b) careerplanning; c) age-appropriate HIV risk reduction materials, provided in schools or clinics); d) mentorship on potential saving goals. |
| *Mentorship* | Each child had a near peer mentor (undergraduate university student) who visited with them monthly for the duration of the intervention. Mentors followed the SUUBI-Uganda intervention manual [37,70] centered around Resilience theory [71]. Mentorship was intended to help children overcome a variety of challenges they face. |
| *Child Development Accounts (CDAs)* | Both treatment arms received a matched CDA held in the child's name in a financial institution. Four national banks operating in the study area: Centenary Bank, DFCU, Stanbic, and Kakuuto microfinance hosted the CDAs. Any of the child's family members, relatives, or friends were allowed and indeed encouraged to contribute towards the CDA. The account was then matched by the studies [33]. Consistent with institutional theory [72,73], and to: 1) examine the impact of saving match rates on performance; and 2) evaluate the cost-effectiveness of alternative match rates, we used two varying rates: *Bridges* (1:1 match rate) and *Bridges PLUS* (1:2 match rate). A monthly savings account statement was generated for every child to note their accumulated savings. During the intervention, each child, with their primary caregiver as a co-signer, had access to the money in account (excluding matching funds). In the event of an emergency, they could withdraw their money, but not the matching funds. The matching funds were kept in a separate account from the participants' own savings and only accessed when participants were ready to pay for their saving goal— e.g., school or small business developments. |
| *Family Income Generating Activity (IGA)* | Children in both treatment arms were trained on microenterprise development. They could use part of their matched savings for starting an IGA (raising chickens, goats or a milk-producing cow), intended to benefit the entire family. The IGAs were intended to promote economic stability for the families, and to enable the participating child to continue in school without worries of economic insecurity. |

**Fig 1. Description of bridges economic intervention components.**

**Overview of the *Original Bridges* study (2012–2018).** The overall goal of *the Original Bridges* study was to evaluate the efficacy and cost-effectiveness of the *Bridges* intervention (detailed below) for YPoAIDS. The study recruited 1383 school-going adolescents (average age 12.7 years). Participants were included if they met the following inclusion criteria: 1) an AIDS-orphaned child/adolescent, defined as one who had lost one or both parents to HIV/ AIDS; 2) enrolled in school; 3) in the last two years of primary school; 4) between ages 10 to 16 years; and 5) living within a family (broadly defined and not an institution; institutions have different familial needs). Participants were ineligible if they were not able to comprehend the study procedures as assessed during the informed consent process. Participants were recruited from 48 primary schools in the greater Masaka Region of Uganda (Masaka, Rakai, Kyotera, Lwengo and Kalungu, five districts hardest hit by HIV/AIDS—prevalence 11.7% vs. 5.4% national average) [69]. Using a cluster-randomized controlled trial (RCT), 48 primary schools were randomly assigned to three study arms (n = 16 schools in each arm): 1) control arm receiving usual care, 2) the *Bridges* arm receiving a Child Development Account (CDA—see description in Fig 1) with a 1:1 match, financial education, and mentorship), and the *Bridges PLUS* arm receiving a CDA with a 2:1 match, financial education, and mentorship. The only

difference between the two treatment arms was the match rate: 1:1 vs. 1:2. The match rate was specifically varied to address the incentivizing and cost-effectiveness question. As indicated in Fig 1 below, both *Bridges* and *Bridges PLUS* included the following intervention components that had been tested in two earlier studies [31–37].

**Progress of the *Original Bridges* study.** *Economic stability.* The study filled important knowledge gaps on the effect of savings- led EEI on short-term economic stability. We found that several measures of material well-being improved among participants in the treatment condition [74]. Specifically, *Bridges PLUS* arm participants had significantly lower odds of having only a few sets of clothes at both the 12-month (OR = 0.49, 95% CI = 0.24, 0.97) and 24-month follow- ups (OR = 0.36; 95% CI = 0.15, 0.85) compared to participants in the control group. In addition, *Bridges* arm participants had significantly lower odds of having no blanket at both the 12-month (OR = 0.33; 95% CI = 0.12, 0.89) and 24- month follow-ups (OR = 0.19; 95% CI = 0.06, 0.64) compared to participants in the control group [71]. In addition, the intervention increased the odds of owning a family business and the levels of asset holding. Families in both *Bridges* and *Bridges PLUS* arms had significantly higher odds of owning a small business at the 24-month follow-up (for *Bridges*: OR = 2.28; 95% CI = 1.05, 4.95; for *Bridges PLUS*: OR = 2.95; 95% CI = 1.38, 6.29) than families in the control arm. Both treatment arms reported a higher increase in their levels of asset possession from baseline to 12 months than the average increase in the control arm [10,74].

*Mental health functioning.* At 24-months post intervention initiation we found a significant unmediated effect of the intervention on children's mental health (B = −0.59; 95% CI = 0.93, −0.25; p < 0.001; β = −0.33). Moreover, the results suggest that participation in the intervention reduced child poverty at the 12-month follow-up, which in turn improved latent mental health outcome at 24 months (B = −0.14; 95% CI: −0.29, −0.01; p < 0.06; β = −0.08). In addition, at 36 and 48-months, mental health of children in the treatment arms improved by 0.13 and 0.16 standard deviation points correspondingly with no evidence of mediation [75].

*Sexual Health, HIV Knowledge/Attitudes and Sexual Risk-Taking.* We found that in the short-run, adolescents in the intervention arms were more likely than control arm participants to report increased scores on HIV knowledge (*b* = .86, 95% *CI* = .47–1.3, *p* ≤ .001), better scores on desired HIV-related beliefs (*b* = .29, 95% *CI* = .06, .52, *p* ≤ .01), and better scores on HIV prevention attitudes (*b* = .76, 95%*CI* = .16, 1.4, *p* ≤ .01) [76–78]. We however acknowledge the potential self-report bias inherent in sensitive lines of inquiry that may have biased these reported short-term results.

*Educational Achievement.* At 24-months post-intervention, we found a 27-percentage point difference, with participants in the intervention arms exhibiting higher likelihoods of schooling at least 90% of the time over a 2-year period. Regarding other educational outcomes, on average, adolescents receiving both interventions showed lower dropout rates, a higher likelihood to take the Primary Leaving Examinations (PLE) and score higher on these PLEs [12,78].

*Cost-effectiveness.* The *Original Bridges* was the first study, to our knowledge, to examine the cost- effectiveness of a savings-led economic intervention based on matched savings accounts in SSA. We found that at 24-months post intervention, adolescents in both intervention groups showed better health, mental health, self-concept, self- efficacy, and HIV knowledge compared to those in the control condition [78]. Our analyses at 48-months indicated that intervention effects were sustained, and higher incentive matching positively impacted more outcomes than lower matching. Specifically, at 48-months, we found that *Bridges PLUS* significantly improved self-rated health (0.25, 95% CI 0.06–0.43), HIV knowledge (0.21, 95% CI 0.01–0.41), self-concept (0.26, 95% CI 0.09–0.44), and adherence self-efficacy (0.26, 95% CI 0.09–0.43), and lowered hopelessness (-0.28, 95% CI -0.43–0.125); whereas *Bridges* significantly improved only self-rated health (0.26, 95% CI 0.08–0.43) and HIV knowledge (0.22,

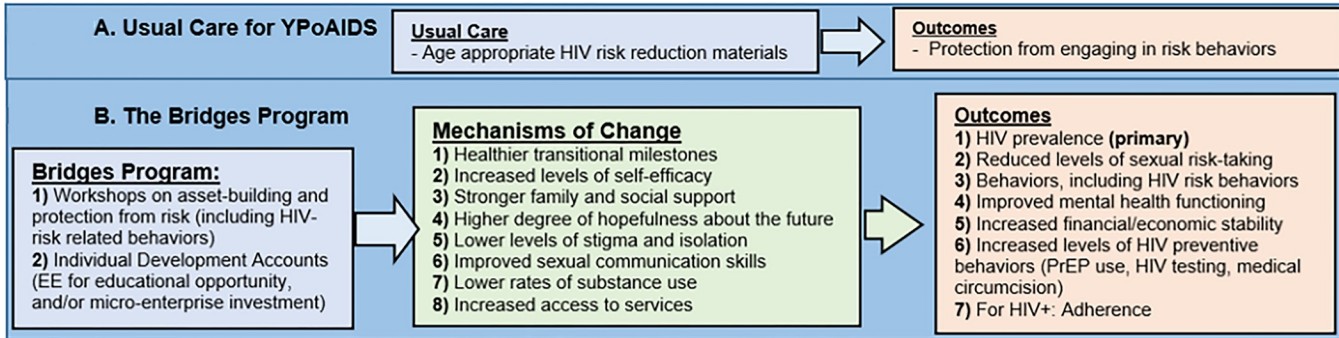

**Fig 2.** A. Usual care for YPoAIDS. B. The bridges program.

95% CI 0.05–0.39). Incremental Cost-Effectiveness Ratios (ICERs) ranged from $224 for hopelessness to $298 for HIV knowledge [10,11,78].

Overall, the *Original Bridges* data have made significant contributions to the literature on the short-term effects of savings-led EI interventions for YPoAIDS. However, we do not know for how long the observed short-term behaviors will be sustained, especially as participants transition through adolescence into young adulthood.

## Methods

### The current study

The *Bridges-R2* study will examine the longitudinal developmental and health outcomes [(e.g., sexual risk-taking behaviors, HIV antiretroviral therapy adherence, PrEP use (for those who are HIV negative)], optimizing health outcomes along the HIV care continuum, and potential mechanisms of the effect of protective health behaviors) of the *Bridges* cohort. Specifically, based on emerging adulthood [13,14] and asset theory [59,60], we expect that the Bridges program would impact youth developmental and health outcomes by reducing the vulnerabilities associated with orphanhood which interfere with the normal development, including reducing stigma and discrimination, improving family support, social support, hopefulness, general self-efficacy, and improve sexual communication skills, and overall facilitate healthier transitions into adulthood (see Fig 2A and 2B). The data will be collected at baseline following reconsenting, 12, and 24-months.

### Inclusion criteria

All *Original Bridges* participants (N = 1383), regardless of number of completed follow-up interviews, will be eligible for *Bridges-R2*. The youngest participant will be 18+ years of age.

We will rely on the *Original Bridges* participants completed future contact forms, the community trust, including with the schools, and Masaka Diocese parish priests for re-consenting. The future contact forms completed by all enrolled participants, provide details on their last known home address, and a list of people who may know their whereabouts, even when they move. Through our field office located in Masaka, we recently confirmed our ability to track and reconsent the majority (>80%) of the 1249 participants from Wave 5 (N = 999).

### Sample tracking, retention and attrition

The *Original Bridges* study (2012–2018) had an attrition rate of 9.7%, leaving a total of 1249 participants at the end of the study (see Table 1). The reported low attrition rates, and the

**Table 1. Bridges retention rates.**

| Baseline | Wave 2 | Wave 3 | Wave 4 | Wave 5 |
|---|---|---|---|---|
| N = 1383 | N = 1321 | N = 1221 | N = 1228 | N = 1249 |
| Retention | 95.5% | 88.3% | 88.8% | 90.3% |

team's ability to easily track and reengage participants over longer periods of time are attributable to: a) the study team having built trust with community members over the 17+ years; b) we have a fully-fledged field office in the Masaka region allowing for easy engagement between the study team and the local collaborators; c) the team has a well-established tracking system, with at least six NIH-funded studies resulting in very low attrition rates; d) unlike open cohort studies, the *Bridges* study participants have been engaged with the study team for an EEI as opposed to an open surveillance cohort. In such cases, the attrition rate tends to be low because of the active engagement and investment in the participants from the very start of the study.

In addition, the *Bridges* study took place in a highly stable region of Uganda where geographical moves among school-going adolescents are rare. The *Bridges-R2* will be conducted in the same region. During the re-consenting process, we will ask participants to update their future contact details, currently on file, with updated telephone number(s) (if they have one: 95% of participants report having access to a phone in the household), and an updated list of names and contact information of three people who will always know how to reach them. Participants will be reminded that if we contact the people listed, we will not discuss any details about their study participation. Like we have done in the past, we will use these records solely to help track their location and only if we have lost contact. We have effectively used these procedures in our previous research studies, as well as our most recent competitive renewal study with YLHIV [79]. Thus, we conservatively expect to re-consent at least 80% of the original sample; and we expect attrition from the reconsented participants to be no more than 20% by the end of the follow-up period. Power analyses (described in the statistical power analysis section) indicate our study is powered to detect medium standardized effects with attrition levels up to 20%, so the impact of attrition on power to test our proposed primary hypotheses for specific aims 1 and 2 should be minimal.

## Statistical power analysis

Minimum detectable effect sizes were estimated using NCSS PASS [80] for the primary analyses proposed to fulfill primary specific aims 1 and 2. Assuming a 20% attrition rate between the end of the *Original Bridges study* (July 2017; N = 1249) and the *Bridges-R2* study (July 2022), our starting N for *Bridges-R2* will be 999 participants. Following a maximum attrition rate of 20%, we anticipate a minimum of 799 participants available at the final measurement wave. We further assume alpha = .017 and power = .80. For specific aim 1, we assumed a control group HIV prevalence of 19.7% based on published data for 15–24-year-old youth from the same geographic region [81] and an intraclass correlation (ICC) of .12 based on our current *Bridges* study data. Assuming these inputs, a two-level multilevel model would be able to detect a raw proportion difference of 15.3%, which corresponds to a standardized difference $h$ = .499, which is a medium minimum detectable effect size [82]. For specific aim 2, we computed the minimum standardized mean difference $d$ for a continuous mechanism of change outcome in a three-level multilevel model assuming the same inputs as above and a maximum within-participant ICC of .70 for the continuous change mechanisms in the *Bridges* study. Under these conditions, $d$ = .462, which is a medium minimum detectable standardized mean difference [82].

### Ethics and consent

All study procedures were approved by the Washington University in St. Louis Review Board (IRB # 201703104) and by in-country local IRBs in Uganda: Uganda Virus Research Institute-UVRI (GC/127/900), and Uganda National Council of Science and Technology -UNCST (SS2586). Amendments to the study protocol will be submitted for approval to the above-mentioned IRBs. Participation in the *Bridges-R2 study* will be voluntary. Written informed consent will be obtained from all participants. This will be done prior to assessment. In the consent form, it will clearly be stated that the participant can: withdraw from the study at any time, for any reason, with no explanation, and would not be penalized in any way; refuse to answer any questions at any time; review any materials and request that we erase any of their responses; make inquiries and address complaints to Secretary of the Ethics Committee at UVRI, UNCST, and Washington University in St. Louis. Participants will also be told of the potential risks and benefits of participating in the study. Each participant will receive a copy of the signed consent form.

### Data collection

**Quantitative assessments.** Assessments will occur at baseline following reconsenting, 12, and 24-months and will take place at the ICHAD's field offices in Masaka or satellite sites (including health clinics) for study collaborators: Mildmay or Reach the Youth (RTY) Uganda, with each lasting about 60–90 minutes. Interviews will be conducted by interviewers fluent in Luganda and English depending on participant's English proficiency. A list of standardized instruments that will be included in the main statistical analyses are outlined in Table 2. All measures used have been pre-tested and made culturally appropriate to the Ugandan context in our ongoing Suubi+Adherence-R2 study [79]. Assessments capturing non-sensitive information, for example, sociodemographic information, mental health functioning, psychosocial well-being and social support are administered orally but recorded using Qualtrics–a computer assisted survey implementation program.

**Biological assay.** Mildmay, with internationally accredited laboratories by SANAS (https://www.sanas.co.za) will be responsible for collection of all biomarkers, participants' counseling, notification, referral for treatment, follow-up and monitoring procedures for biological testing for HIV and viral load. Biological assay will be collected at T1 following reconsenting (T1-RC), and at 12 and 24-month post-reconsenting by Mildmay Medical Staff. HIV test results will be communicated to the participants by Mildmay.

**Qualitative data collection.** Semi-structured qualitative interviews will be conducted at one time point (T1) immediately following the completion of quantitative assessments at T1-RC. The interviews will focus on participants' experiences with the *Original Bridges* study, plus an exploration of: 1) key multi-level factors that may have affected participants' decision-making and behaviors around savings and sexual risk taking; 2) social transitions and their impact on sexual risk taking, saving, and mental health; and 3) perceptions of costs and rewards of saving and safe sexual behaviors. We will explore whether these experiences and perceptions differ between participants in the control and intervention arms. For participants in the two treatment arms only, we will address their experiences with the intervention and its specific components; and key multi-level factors (individual, family, and programmatic) that may have impacted their participation in the intervention. Using quantitative data obtained at T1-RC, a stratified purposeful sampling strategy [103] will be used to select a total of 60 participants (n = 20 per study arm) from the highest and lowest quartiles on two key outcomes (sexual risk-taking and mental health). The selected 60 participants will be invited to participate in the semi-structured interviews. These numbers will be sufficient to achieve theoretical

**Table 2.** *Bridges-R2* study measures.

| Variable | Measurement | Timepoint |
|---|---|---|
| **Demographics** | | |
| Socio-demographics: Age; sex (assigned at birth), orphanhood status; socioeconomic status; family composition | | **T1-RC, 12, 24 |
| **Moderators** | | |
| Rural/semi-urban; exposure to outside HIV-related programs; household income; asset accumulation | | **T1-RC, 12, 24 |
| **Potential Mechanisms of Change** | | |
| Self-efficacy | Adapted Tennessee Self-Concept Scale (TSC-2) [83] | **T1-RC, 12, 24 |
| Social Support | Social Support Behaviors Scale (SS-B) [84]<br>Family Cohesion Scale [85] | **T1-RC, 12, 24 |
| Hopelessness | Beck Hopelessness Scale [86] | **T1-RC, 12, 24 |
| Stigma and Isolation | Adapted Social Impact Scale [87] | **T1-RC, 12, 24 |
| Self-esteem | Rosenberg Self-Esteem Scale [88] | **T1-RC, 12, 24 |
| Transition milestones (residential independence, education, employment, parenthood, prosocial transition, relationship/marital status) | Relationship assessment scale [89,90]<br>Additional prosocial activities, Demographics, Work history interview | **T1-RC, 12, 24 |
| Access to services | Treatment Services Review (TSR) adult versions [91,92] | **T1-RC, 24 |
| Substance misuse | Alcohol, Smoking, Substance Involvement Screening Test [93] | **T1-RC, 12, 24 |
| **Primary Outcome** | | |
| HIV prevalence | Biological assay | **T1-RC, 12, 24 |
| **Secondary Outcomes** | | |
| Sexual risks | Timeline Follow Back (TFLB) [94,95], Adapted Youth Aids Prevention Project (used in Suubi & CHAMP) [96] | **T1-RC, 12, 24 |
| Mental Health Functioning | Center for Epidemiological Studies-Depression Scale (CES- D) [97], PTSD Checklist [98], Brief symptom inventory [99] | **T1-RC, 12, 24 |
| Financial/economic stability (Savings and asset) | Verifiable Bank statements, MIS IDA [100,101]; Self-Reports | **T1-RC, 12, 24 |
| Adherence (for YLHIV) | Wilson self-report Questionnaire [102], viral load | **T1-RC, 12, 24 |
| HIV testing | Self-report, verified by clinic records | **T1-RC, 12, 24 |
| PrEP use (for HIV negative participants) | Self-report, verified by clinic records | **T1-RC, 12, 24 |
| Medical Male Circumcision (for boys) | Self-report, verified by clinic records | **T1-RC, 12, 24 |

** T1-RC–Time 1 immediately following re-consenting.

saturation [103–105]. Moreover, the sampling strategy will ensure that participants with varying experiences within the same study arm are represented; and allow for identification of common patterns and variations in participants' experiences (e.g., unique vulnerabilities and protective factors). The semi-structured qualitative interviews will be conducted in English or Luganda based on participants' preference. Questions will be translated (English to Luganda) and back-translated by two proficient team members. Questions will be reviewed by two native speakers to make sure that they sound natural and conversational, and revised accordingly. Each interview will last about 60 minutes and will be audio-taped.

## Data analysis plan

**Data quality assurance, initial analyses, and missing data.**   We will use MIS IDA Q [101] to check for data-entry errors, missing values, and accounting inconsistencies. Frequency tables for all variables and measures of central tendency and variability for continuous variables will characterize the sample overall and by randomization group. We will address incomplete data with direct maximum likelihood (ML) and multiple imputation (MI) [106] because they make the relatively mild assumption that incomplete data arise from a conditionally missing-at-random (MAR) mechanism [107]. Auxiliary variables will be included to help meet the MAR assumption [108,109] and sensitivity analyses will be conducted with pattern-based MI [110,111] to assess the robustness of the MAR assumption. [112]. SAS [113] and Mplus [114] will be used to perform the proposed analyses.

**Time points used in proposed inferential analyses.**   To maximize the rigor of the analyses; to assure alignment of biomarker data (only measured in the *Bridges-R2*) and our remaining outcomes; and to avoid reuse of *Original Bridges* study outcomes (already examined for shorter-term intervention efficacy) and biased examination of longer-term intervention efficacy, all proposed primary inferential analyses and most of the proposed secondary inferential analyses listed will use only *Bridges-R2* data. The exception is longitudinal developmental trajectory analyses proposed to explore associations between changes in time-varying covariates and mechanisms of change with outcomes measured across both studies.

**Primary analyses for Aim 1.**   We hypothesize that: H1a. Relative to the control group, participants in the *Bridges* intervention group (1:1 savings incentive match rate) will have a lower odds of HIV infection at the final measurement point; H1b. Relative to the control group, participants in the *Bridges PLUS* intervention group (1:2 savings incentive match rate) will have a lower odds of HIV infection at the final measurement point; and H1c. Relative to the *Bridges* intervention group, participants in the *Bridges* PLUS intervention group will have a lower odds of HIV infection at the final measurement point. To test H1a-H1c, we will fit a two-level generalized linear mixed model (GLMM) with fixed effects for study arm, time, and their interaction. We will use random intercepts for school ID to account for clustering of persons within schools. Reflecting the binary HIV status outcome, a binomial distribution and logit link will be used. To test H1a-H1c we will perform three pairwise planned comparisons. Alpha ($\alpha$) will be set at .05/3 = .017 for these three comparisons to maintain a nominal $\alpha$ = .05. We will adjust for orphanhood status, the only covariate where imbalance was detected in our *Original Bridges* data. It is important to note that previously, deaths have been very few; however, if numbers of deaths become non-trivial, we will extend our proposed GLMMs to jointly model survival and the odds of HIV infection simultaneously [115].

**Secondary exploratory analyses for Aim 1.**   Exploratory analyses will explore for all participants the superiority of a) *Bridges* to control, b) *Bridges PLUS* to control, and c) *Bridges PLUS* to *Bridges* in lowering the odds of engaging in sexual risk taking, the odds of problem alcohol and substance use behaviors, and increasing mean levels of mental health functioning among study participants. Subgroup analyses will perform the same comparisons for HIV testing and PrEP use for the subset of participants not living with HIV; a parallel set of analyses will perform the same comparisons for ART adherence and HIV care engagement for youth living with HIV. The same GLMM approach described above will be extended to three levels to model the odds of binary outcomes across time using the same fixed effects (study arm, time, and study arm-by-time), a random effect for the school level (random intercepts), and adding random effects for the person level (random intercepts, random slopes, and their covariance). For the continuous mental health exploratory outcomes, we will fit linear mixed models (LMMs).

To maximize rigor, the assumptions of normality and constant variance of residuals in LMMs will be evaluated by examining histograms of the residuals and scatter plots of predicted values-by-Cholesky-scaled residuals, respectively. Transformations of outcomes will be utilized as needed to improve data conformance with model assumptions. Inferences for models whose residual statistics still do not fully meet assumptions following transformations will be generated via robust heteroskedastic-consistent estimators [116]. Group comparisons will be performed via three pairwise time-averaged comparisons across repeated measurements. Alpha ($\alpha$) will be set at .05/3 = .017 to maintain a nominal $\alpha$ = .05 per outcome. Any additional post-hoc comparisons (e.g., paired comparisons of groups at each time point) will maintain nominal $\alpha$ = .05 through the use of simulation-based stepdown multiple comparison methods [117]. All analyses will include outlier and influential case screening via computation of Cook's D, DFBetas, and likelihood displacement statistics. If outliers are found, results will be reported with and without outliers included [118,119].

**Primary analyses for Aim 2.** We hypothesize that: H2a. Relative to the control group, participants in the *Bridges* intervention group (1:1 savings incentive match rate) will have higher mean levels of economic stability, positive youth transitions, and social support during the *Bridges-R2* study; H2b. Relative to the control group, participants in the *Bridges PLUS* intervention group (1:2 savings incentive match rate) will have higher mean levels of economic stability, positive youth transitions and social support in the Bridges-R2 study, and; H2c. Relative to the *Bridges* intervention group, participants during the *Bridges PLUS* intervention group will have higher mean levels of economic stability, positive youth transitions and social support. The same three-level linear mixed model (LMM) approach as described in the previous paragraph will be used to test these hypotheses. Group comparisons will be performed via three pairwise time-averaged comparisons across repeated measurements. $\alpha$ will be set at .05/3 = .017 to maintain nominal $\alpha$ = .05 per outcome.

## Secondary exploratory analyses for Aim 2

**Exploring mediation and moderation.** Exploratory analyses will explore whether the potential mechanisms of change included in Table 2 mediate the relationships between intervention group assignment and CR-HIV, economic stability, positive youth transitions and social support, and whether geographic location, exposure to other HIV/STI prevention programs, household income and asset accumulation moderate those associations. Analyses will be conducted using principles of structural equation modelling (SEM) and causal inference [120]. M*plus* will be used to fit mediation and moderation models because it can adjust standard errors for clustering of participants within schools and because it unites causal inference and SEM-based analyses in the same software platform [121].

**Exploring trajectories of change over the full developmental period captured by the *Bridges* project.** Moderator, mediator, and sexual behavior and mental health outcome data will be available at 8 time points from the start of *Original Bridges* through the end of *Bridges-R2*. We will initially explore trajectories of these variables by generating spaghetti plots of participants' individual trajectories with intervention group averages overlaid to visualize group average (i.e., mean) levels of change and inter-participant variability. Next, we will fit mixed effects growth models to investigate whether change over time can be quantified using low-dimensional parametric functions (e.g., linear, quadratic) of explanatory variables. Finally, we will employ the highly flexible time-varying effect modeling (TVEM) to go beyond simple parametric functions to describe the form of changes in variables over time. TVEM allows the effects of covariates on outcome trends to vary over time non-parametrically [122,123]. TVEMs will be fitted using the % weighted TVEM SAS macro produced by the Penn State Methodology Center, a leader in developing longitudinal analysis methods and software.

**Sex as a biological variable.** All previously described analyses will be repeated with models extended to include sex assigned at birth as a moderator to examine whether effects vary by participant sex. For male participants, we will perform additional exploratory analyses at each time point using a three-level GLMM to explore whether medical male circumcision differs by intervention group.

**Qualitative data analysis for Aim 3.** All interviews will be transcribed and translated from Luganda into English. Transcripts will be compared with digital records to ensure transcription accuracy. All transcripts will be uploaded to QSR NVivo12 analytic software for data analysis. Interview transcripts will be reviewed by the research team to develop a broad understanding of content as it relates to the project's specific aim and to identify topics of discussion and observation. During this step, as well as during subsequent steps, analytic memos will be written to further develop categories, themes, and subthemes, and to integrate the ideas that emerge from the data [124]. Using analytic induction techniques [125], transcripts will be read multiple times for initial coding by the research team. For initial coding, randomly selected 10 interview transcripts will be read multiple times and independently coded by the project investigators using a priori (i.e., from the interview guide) or emergent themes (also known as open coding) [124]. Broader themes/ categories will be broken down into smaller, more specific units until no further subcategory is necessary. Qualitative findings will also shed light into potential mediators and moderators for sexual risk-taking and mental health.

The codes and definition boundaries of specific codes (inclusion and exclusion criteria for assigning a specific code) [125] will be discussed during research team meetings to create the final codebook. Each text will be independently coded by two investigators using the codebook. Inter-coder reliability will be established. A level of agreement from 66 to 97% depending on level of coding (general, intermediate, specific), indicates good reliability in qualitative research [126]. Disagreements will be resolved through discussion during team meetings. The secondary analysis will focus on comparing and contrasting themes and categories within and across groups to identify similarities and differences. To further ensure rigor, study results will be presented to study participants who participated in the interviews, enabling them to provide comments on results and suggest modifications or additional avenues of investigation when possible (member checking) [127]. An audit trail of data collected as well as memos and minutes of team meetings will be kept throughout the study [127].

**Cost-Benefit Analysis (CBA) for Aim 4.** The *Original Bridges* study used cost-effectiveness analysis (CEA) a tool for evaluating the costs of producing a given level of outcome [11]. CEA allows users to compare different approaches (*Bridges* or *Bridges PLUS*) for achieving these outcomes and to identify more efficient ones—those with a smaller Incremental Cost-Effectiveness Ratio (ICER), the $/outcome (in USD or local currency, UGX), relative to a status quo ("usual care"). CBA in this study builds upon the CEA by valuing (monetizing) all outcomes of *Bridges* in USD (and UGX), facilitating critical assessments and comparisons for sustainability and scalability. Building on the cost data in our previous CEAs, and continued evaluation in *Bridges-R2*, we will use CBA methods [128,129] to: 1) value gains observed since the start of *Bridges and Bridges PLUS* and 2) develop lifecycle economic models of projected future earnings, health trajectories, and social improvements. In Aim 1, the net change in HIV prevalence will be valued (prospectively) as the survival-adjusted lifecycle costs of HIV in DALYs from the GBD [130,131] converted to UGX & USD [132]. We will use a similar approach for the difference in mental health and substance misuse, secondary outcomes in Aim 1; that is, the change in rates will be measured in DALYs which are then monetized. If rates decrease, *costs avoided* are a benefit to *Bridges-R2*, and if rates increase, costs incurred reduce the net benefit. For Aim 2, the difference in viral suppression, PrEP use, and circumcision has both a cost (increased health care services) and a benefit (reduced risk of adverse

health outcomes). The health benefits of viral suppression, PrEP use, and circumcision will be taken from the literature and valued in DALYs converted to UGX and USD as above. For economic stability, the present timing as YPoAIDS in *Bridges* transition into young adulthood is ideal: models will be based on actual earnings and assets. In pure economic value, we anticipate that the gains in earnings via improved human capital and financial behaviors will be the largest direct economic benefit. Following economic evaluation standards [133], we will report the CBA from 3 perspectives (participants, funders, and society), discount benefits and costs to their present value, and conduct sensitivity analyses around valuation parameters, modeling of future effects, and discounting.

**Data integration.** The study uses an explanatory sequential design [134]. Qualitative and quantitative data analyses will be performed separately. Findings will be integrated at the interpretation and discussion stages. Conclusions and inferences will be synthesized for a more contextualized and thorough understanding of intervention impact. Data integration will serve two purposes: 1) Complementarity and 2) Expansion [135,136]. Qualitative and quantitative findings will be connected, with the former providing explanations and context for findings produced by the latter. Qualitative data will potentially provide further context to the EI impact on primary outcomes and mediators over time with questions focused on multi-level factors impacting decision-making and behavior.

## Monitoring and responding to adverse events

We will identify, manage, and document events and psychological distress reactions that actually occur during the study period. These events may be identified by project staff or reported by participants. All study personnel, including data collectors and facilitators will receive extensive training on how to identify verbal and non-verbal signs that may indicate psychological distress and adverse events. They will also be trained on how to support distressed participants and to offer referrals to local clinics/ hospitals if necessary. The in-country teams at RTY-Uganda, Makerere University, Mildmay-Uganda and Washington University's ICHAD field offices are knowledgeable of resources available to participants in the study region. If the need arises, Research Assistants will make appropriate referrals for basic and enhanced services.

Reporting of adverse events will occur according to a project protocol. For this study, safety and monitoring will be overseen by three MPIs (Drs. Ssewamala, Sensoy Bahar and Nabunya (Washington University in St. Louis), and in-country PI (Dr. Nakasujja based at Makerere University). This group is expected to have weekly telephone conference calls (using zoom). In the case of an adverse event, staff will immediately notify the MPIs. Any presence of a possible unanticipated adverse event will be immediately reported and brought to the attention of the Washington University Institutional Review Board (along with the Ethics Committee at UVRI and UNCST). The IRBs will determine whether it is appropriate to stop the study protocol temporarily or will provide suggestions and/or modifications to the study procedures. Possible modifications may include adding new risks to the consent form and re-consenting all study participants.

Preliminary outcomes data will be examined quarterly by the research team. If preliminary outcome data indicates harmful impact of the program to the study participants, Washington University IRB committee, as well as the Ethics Committee at UVRI and UNCST will be notified.

## Data management and integrity to protect confidentiality

To protect the integrity of the participants' data, the following procedures will be followed. First, the data collected from the study participants will be used only for the purpose of

research. All data will be kept confidential. We will not share any information or answers we get from the participants with their parents, peers, relatives, colleagues etc. Second, all participants in the study will be assigned a code number by the research staff. This code number is used on all information collected from participants, including questionnaires. Since the study is a longitudinal one, we will maintain lists of participants with links between identifying information and code numbers. Only the MPIs, in-country PI, Project Coordinator and Data Manager will have access to these lists, which are kept in locked files. Other study personnel will have access on an as needed basis to individual participants' names and code numbers in order to adequately perform their duties. For example, interviewers must label the questionnaires with the correct code number of the participant whom they are interviewing.

All personnel will be required to complete certain levels of training before they are granted access to this identifying information. They will have complete the Human Subjects Training sponsored by the National Institute of Mental Health, which complies with federal guidelines delineated in 45 CFR Part 46. Personnel will also sign confidentiality statements that specify that if the participants' confidentiality is breached unintentionally that personnel will follow the procedures for reporting this breach to the MPIs. The confidentiality statements will state that unintentional or deliberate violations of participants' confidentiality may result in demotion or termination depending upon the severity of the event. The project personnel will also participate in training with the MPIs and the in-country PI, regarding data safety, confidentiality of participants, limits of confidentiality, and proper administration of the study protocol. All hard copies of data will be stored in locked cabinets to which only the MPIs, in-country PI, Project Coordinator and Data Manager have access.

After completion of an interview with a study participant, data with code numbers will be placed in a separate locked file cabinet while waiting for entry. Once data is entered into computer files and password protected, only the MPIs, In-country PI, Project Coordinator, Data Manager, plus data entry clerk (on as needed-basis) will have access to these files. All requests, current and future, to use the data will be reviewed by the MPIs. Any data files that are provided to other individuals will be stripped of identifiers and contain only code numbers so that data across multiple assessment waves can be matched. Participants will be notified of the above procedures during the informed consent process.

## Discussion

Growing evidence suggest that economic strengthening interventions, including conditional or unconditional cash transfers, microfinance and savings, employment and vocational training, can reduce behaviors that increases HIV risk, particularly among adolescents and young people [137,138]. Similarly, studies have documented the impact of household economic strengthening on HIV related outcomes of YLHIV, including retention in HIV care and treatment adherence [139,140]. Moreover, while there are ground-breaking prospective cohort studies in low-resource communities examining the longitudinal developmental and transition milestones for orphaned youth and YLHIV [20,141–144] as well as open population-based cohort studies [145], none of the studies we are aware of examined the long-term (beyond a 5-year period) impact of an economic intervention that may be a protective factor for YPoAIDS, including those living with YLHIV. Findings from this research may provide an unprecedented opportunity to examine the long-term impact of an economic intervention on health-risk behaviors of YPoAIDS as they transition through young adulthood.

The study is innovative in several ways. First, to date, HIV prevention, care and support intervention efforts in SSA communities have primarily been "transported" from outside the region, mainly from the global north [146–148]. We know little about how economic

interventions aimed at YPoAIDS, including YLHIV, would fare over time, and as they are scaled up in low-resource contexts. Yet, such knowledge is imperative as we build and implement a policy agenda for care and support for this population. For example, in line with Uganda's multi-sectoral HIV and AIDS control approach, the Ministry of Education and Sports established the HIV and AIDS structure and activities focused on eliminating new HIV infections within the Education and Sports Sector (ESS). As such, findings from our study may contribute research informed evidence to the goals of this section including those related the reduction in the number of persons engaged in HIV high risk behaviors, increase in access to prevention, care, treatment and social support services, as well as strengthening the capacity of ESS institutions to plan, implement, coordinate, monitor and evaluate their HIV prevention programs. Moreover, it is critical to identify efficacious and potentially replicable interventions for the global south's existing infrastructure, with proven longer-term effects. Second, to our knowledge, *Bridges-R2* would be the first to examine the long-term (beyond a 5-year period) impact of an economic intervention that may be a protective factor for YPoAIDS. Third, in the *Original Bridges*, we relied on the most age-appropriate self-reports regarding sexual-risk behaviors because participants were young (average age at enrollment was 12.7 years). In the *Bridges-R2*, we incorporate biomedical measures, including HIV testing and Viral load (for YLHIV). Finally, the study uses developmental and emerging adult theoretical frameworks [13,14] to identify pathways for successful and problematic transitions for YPoAIDS in Uganda, determining strategic points of intervention. By knowing the key pathways that enable poor youth to successfully overcome the consequences of HIV/AIDS and chronic poverty, practitioners and policymakers can make informed program choices regarding investment in the most cost-effective programs.

The research team will leverage the current Ugandan government policy guidelines regarding youth empowerment contained in Vision 2040 framework to maximize dissemination of study findings. In addition to journal publications, the research team uses a range of strategies, including annual reports, monthly newsletters, and policy briefs, as well as annual stakeholders meetings and an annual global conference that engages researchers, NGOs, and government officials from the U.S. and across SSA, to disseminate research findings.

## Conclusion

As youth, especially those orphaned by HIV/AIDS, transition into young adulthood, they are faced with limited social support and opportunities for education and employment, which elevates their vulnerability to poverty, poor mental health functioning and other negative life outcomes, including risk-taking behaviors exposing them to HIV/AIDS and poor health outcomes. The *Bridges-R2* study will examine the longitudinal impact of a family-based economic empowerment intervention on the developmental and behavioral health outcomes, and transition milestones for young people orphaned by AIDS in Uganda. Findings from this large longitudinal dataset may contribute to the scientific knowledge on the potential value of investing modest economic resources in poor and vulnerable boys and girls during early adolescence and how these resources may offer long-term protection against known HIV and mental health risks—two public health issues impacting millions of young people in low-resource communities.

## Acknowledgments

We are grateful to our staff and volunteers at the International Center for Child Health and Development (ICHAD) and Reach the Youth (RTY) Uganda for their respective contributions to study implementation. In addition, we are grateful to the participating schools, participating youth and their caregiving families for agreeing to participate in the study.

## Author Contributions

**Conceptualization:** Proscovia Nabunya, Ozge Sensoy Bahar, Fred M. Ssewamala.

**Funding acquisition:** Proscovia Nabunya, Ozge Sensoy Bahar, Fred M. Ssewamala.

**Investigation:** Proscovia Nabunya, Noeline Nakasujja, Fred M. Ssewamala.

**Methodology:** Torsten B. Neilands.

**Project administration:** Ozge Sensoy Bahar, Fred M. Ssewamala.

**Resources:** Proscovia Nabunya, Ozge Sensoy Bahar, Fred M. Ssewamala.

**Supervision:** Proscovia Nabunya, Flavia Namuwonge, Fred M. Ssewamala.

**Writing – original draft:** Proscovia Nabunya, Ozge Sensoy Bahar, Fred M. Ssewamala.

**Writing – review & editing:** Proscovia Nabunya, Ozge Sensoy Bahar, Torsten B. Neilands, Noeline Nakasujja, Phionah Namatovu, Flavia Namuwonge, Abel Mwebembezi, Fred M. Ssewamala.

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
