## [Decision Letter · Decision Letter 0]

5 Aug 2022

PONE-D-22-11909

Bridges-Round 2: A study protocol to examine the longitudinal HIV risk prevention and care continuum outcomes among orphaned youth transitioning to young adulthood

PLOS ONE

Dear Dr. Nabunya,

Thank you for submitting your manuscript to PLOS ONE. After careful consideration, we feel that it has merit but does not fully meet PLOS ONE’s publication criteria as it currently stands. Therefore, we invite you to submit a revised version of the manuscript that addresses the points raised during the review process.

The manuscript has been evaluated by two reviewers, and their comments are available below.

The reviewers have requested additional methodological detail, as well as additional contextualization for the study and information regarding the rationale. 

Could you please revise the manuscript to carefully address the concerns raised?

We look forward to receiving your revised manuscript.

Kind regards,

Vanessa Carels

Staff Editor

PLOS ONE

Journal Requirements:

Reviewers' comments:

Reviewer's Responses to Questions

**Comments to the Author**

1. Does the manuscript provide a valid rationale for the proposed study, with clearly identified and justified research questions?

Reviewer #1: Yes

Reviewer #2: Yes

2. Is the protocol technically sound and planned in a manner that will lead to a meaningful outcome and allow testing the stated hypotheses?

Reviewer #1: Yes

Reviewer #2: Partly

3. Is the methodology feasible and described in sufficient detail to allow the work to be replicable?

Reviewer #1: Yes

Reviewer #2: No

4. Have the authors described where all data underlying the findings will be made available when the study is complete?

Reviewer #1: Yes

Reviewer #2: Yes

5. Is the manuscript presented in an intelligible fashion and written in standard English?

Reviewer #1: Yes

Reviewer #2: Yes

6. Review Comments to the Author

You may also provide optional suggestions and comments to authors that they might find helpful in planning their study.

Reviewer #1: Review of PONE-D-22-11909: “Bridges-Round 2: A study protocol to examine the longitudinal HIV risk prevention and care continuum outcomes among orphaned youth transitioning to young adulthood”

Summary: This study proposes a continuation of the ongoing Bridges study to follow participants into emerging adulthood.

Review: This is an important study and there are meaningful contributions that can be made by continuing to follow participants and gather more data. I would like more information regarding payment of participants. Will they be compensated for this study or will additional money be added to their accounts?

Reviewer #2: Review: PONE-D-22-11909

“Bridges-Round 2: A study protocol to examine the longitudinal HIV risk prevention and care continuum outcomes among orphaned youth transitioning to young adulthood.”

Major revision

Summary: This article presents a study protocol to examine longitudinal HIV-orphan outcomes (e.g., sexual risk behavior, mental health, savings) 5-10 years post-intervention, including the control sample; subjects are now young adults.

Overall, this is an exciting follow-up to an economic intervention that targeted orphaned youth (ages 10-16 but see 16 below) who are now young adults. Also laudable is the mixed methods approach, which holds great promise to be informative. Yet, there is no clear presentation of the connection between the theoretical context and the verifiable theory-based measurement-grounded model linked to the proposed hypotheses. There is a reference to the “model of change,” but it is not clearly explicated. Table 2 is confusing, as it has references to secondary but not primary outcomes. It lists moderating and mediating variables without a clear explanation of what causal effects are assumed to be captured and how they will be evaluated. The theoretical context—hypotheses—assessment/measurement triangulation is weak, as presented, and needs much more work and clarity. As it is “packaged” now, it is not replicable/reproducible, which challenges the very assumption behind registering this protocol. It reads as a copy-paste from a grant application; this is not what is needed for a publication as envisioned here.

Minor comments:

1. In the Introduction, line 92 (and elsewhere), the relevance of the study to important policy questions is mentioned. Can you please note more specifically the types of policy/policies you hope to influence with your results?

2. P. 3: Sort out your usage of SSA. Define once at the first usage and stay with the abbreviation.

3. P. 3: “Our findings show short-term success with improving emotional well-being (less 90 depression and hopelessness); reduced self-reported sexual risk-taking behaviors; improved 91 family cohesion, and positive educational outcomes among intervention participants [9-12].” For each outcome, indicate the observed effect size.

4. Pp. 4-5: Please provide more details on what “poverty” means in Uganda

5. P. 5: What do you mean by “compromised mental health”? Please explain.

6. The construct of transitions to adulthood has been defined by five events first proposed in the 1980s in the global north (leaving home, completing school, entering the workforce, forming a romantic partnership, and transitioning to parenthood), as per your citations. Transitions to adulthood as a set of developmental milestones do not appear to have been studied in SSA as yet. Please consider and comment on the possibility that other milestones (related to different roles) may emerge in the course of your research that is more appropriate/relevant to development into adulthood within the Ugandan context.

7. P. 6: Please change the subtitles “theoretical models” to something else; this subtitle does not reflect the content of the section (there are no models).

8. P. 7: It has not yet been explained what a CDA is

9. P. 7: De-green ref 56.

10. P. 11: The sentence “The future 306 contact forms completed by all enrolled participants, provides details on participants 307 whereabouts, even they move.” is agrammatical; please adjust.

11. P. 11: To follow up on the statement, “Through our field office located in Masaka, we recently 308 confirmed our ability to track and reconsent the majority (>80%) of the 1249 participants from 309 Wave 5 (N=999).”, please indicate what kind of power implications are expected with this sample size. There is a relevant discussion on pp. 18-19, but p. 11 should have an overall summary statement on the anticipated impact of attrition.

12. It is stated in the text that “All measures used have been or will be pre-tested and made culturally appropriate to the Ugandan context” (p13). Please specify which have already been pre-tested and made culturally appropriate and which have not. This indicates how much work has to be done prior to the start of the study.

13. P. 13: What is the MTF listed as one of the assessments for the Transition Milestones?

14. P. 14: “These numbers will be sufficient to achieve 394 theoretical saturation [98-100].” What does this sentence mean?

15. P. 16: It has not yet been explained what ICER stands for

16. The recruitment ages for the Original Bridges Project are unclear. On p5 Line 162, it is listed as 10-14 years old; on p7 Line 212, it is listed as 10-16 years old. Please clarify.

7. PLOS authors have the option to publish the peer review history of their article (what does this mean?). If published, this will include your full peer review and any attached files.

Reviewer #1: No

Reviewer #2: No

---

## [Author Response · Author response to Decision Letter 0]

26 Sep 2022

Please see separate document attached for detailed response to reviewer comments.

---

## [Decision Letter · Decision Letter 1]

2 Feb 2023

PONE-D-22-11909R1Bridges-Round 2: A study protocol to examine the longitudinal HIV risk prevention and care continuum outcomes among orphaned youth transitioning to young adulthoodPLOS ONE

Dear Dr. Nabunya,

Thank you for submitting your manuscript to PLOS ONE. After careful consideration, we feel that it has merit but does not fully meet PLOS ONE’s publication criteria as it currently stands. Therefore, we invite you to submit a revised version of the manuscript that addresses the points raised during the review process.

We look forward to receiving your revised manuscript.

Kind regards,

Asrat Genet Amnie, MD, EdD, MPH, MBA

Academic Editor

PLOS ONE

Journal Requirements:

Additional Editor Comments (if provided):

The author (s) is/are urged to make all corrections in sentence structure, tense, style, punctuation, etc. of the manuscript suggested by Reviewer 3, preferably seek professional assistance in the writing of the manuscript . The author(s) is/ are also urged to expand on their introduction and discussion sections using any one or more of the references suggested by Reviewer 4.

Reviewers' comments:

Reviewer's Responses to Questions

**Comments to the Author**

1. Does the manuscript provide a valid rationale for the proposed study, with clearly identified and justified research questions?

Reviewer #1: Yes

Reviewer #3: Yes

Reviewer #4: Partly

2. Is the protocol technically sound and planned in a manner that will lead to a meaningful outcome and allow testing the stated hypotheses?

Reviewer #1: Yes

Reviewer #3: Yes

Reviewer #4: Yes

3. Is the methodology feasible and described in sufficient detail to allow the work to be replicable?

Reviewer #1: Yes

Reviewer #3: Yes

Reviewer #4: Yes

4. Have the authors described where all data underlying the findings will be made available when the study is complete?

Reviewer #1: Yes

Reviewer #3: Yes

Reviewer #4: Yes

5. Is the manuscript presented in an intelligible fashion and written in standard English?

Reviewer #1: Yes

Reviewer #3: No

Reviewer #4: Yes

6. Review Comments to the Author

You may also provide optional suggestions and comments to authors that they might find helpful in planning their study.

Reviewer #1: Summary: This study proposes a continuation of the ongoing Bridges study to follow participants into emerging adulthood.

Review: I approve this manuscript.

Reviewer #3: 1. I suggest rephrasing "potential mechanisms of protective health behaviors" in the abstract to "potential mechanisms of effect of protective health behaviors".

2. Please ensure that any em-dash (e.g. "–but currently unknown") has a corresponding ending dash, or switch to commas (also needed in pairs).

3. When reporting data from the original Bridges intervention, please specify what kind of self-efficacy is being referred to. "Self-efficacy for..." (both in introduction and later, e.g. line 296)

4. There is a typo in line 110 (Brides to the Future)

5. In line 130, I would suggest adding "in" before "all other regions."

6. Put the explanation for the term PrEP next to the first time it is used (similar comment to that made by a previous reviewer regarding "SSA"). Relatedly, if an abbreviation is used (like" YLHIV"), please use it consistently for all future references to that group.

7. When listing out several items, please include "a)" rather than starting the list and then itemizing from b) onwards.

8. It is generally recommended to avoid the use of "don't" in academic writing (see like 303).

9. I would suggest combining the sample tracking and power calculations sections, or at least bringing them closer together.

10. What are the biomarkers for sexual risk-taking? This does not seem clear to me. If HIV and VL only, how are these measuring risk-taking in this sample size? It is unclear how VL changes would be associated with risk-taking...

11. Most of this is written in future tense, however sections on data management and confidentiality switch to present tense.

Reviewer #4: Overall, the proposed study described in this protocol will provide critical insights into the long term effects of economic interventions on YPoAIDS in SSA and makes a meaningful contribution to the literature. The mixed methods design is strong and the authors have clearly laid out the study procedures and analytic plan. I also think they have done a good job of addressing the first reviewer’s comments. I have only a few suggestions for minor revisions that I think will strengthen the protocol:

Minor comments

1.) It would be helpful for the authors to more explicitly describe potential mediators and moderators they plan to examine. Particularly, I think it would be helpful to examine whether intervention effects are moderated by participants’ sexual identities/partner characteristics to see whether the intervention performs differently among young MSM and WSMSM for instance.

2.) I think the section on qualitative analyses on page 20 could benefit from more discussion about how the research team plans to draw comparisons between the intervention and control arms of the study and what they hope to learn from these comparisons. It’s a little less clear than the analytic plan for the quantitative components.

3.) The introduction and discussion would be a bit stronger if the authors include a short 3-4 sentence summary of other economic interventions for HIV-prevention and situate their contribution within that larger literature. The authors do a good job of explaining the unique strengths of their study, but I think it’s less clear how it builds upon an existing and growing set of economic interventions that target structural determinants of HIV across a variety of contexts and populations. Here are a some articles that could be included in this discussion:

Rotheram-Borus, M. J., Lightfoot, M., Kasirye, R., & Desmond, K. (2012). Vocational training with HIV prevention for Ugandan youth. AIDS and Behavior, 16, 1133-1137.

Hill, B. J., Motley, D. N., Rosentel, K., VandeVusse, A., Fuller, C., Bowers, S. M., ... & Garofalo, R. (2022). Employment as HIV Prevention: An Employment Support Intervention for Adolescent Men Who Have Sex With Men and Adolescent Transgender Women of Color. Journal of Acquired Immune Deficiency Syndromes (1999), 91(1), 31.

Souverein, D., Euser, S. M., Ramaiah, R., Rama Narayana Gowda, P., Shekhar Gowda, C., Grootendorst, D. C., ... & Den Boer, J. W. (2013). Reduction in STIs in an empowerment intervention programme for female sex workers in Bangalore, India: the Pragati programme. Global health action, 6(1), 22943.

Rosenberg, M. S., Seavey, B. K., Jules, R., & Kershaw, T. S. (2011). The role of a microfinance program on HIV risk behavior among Haitian women. AIDS and Behavior, 15, 911-918.

Pronyk, P. M., Hargreaves, J. R., Kim, J. C., Morison, L. A., Phetla, G., Watts, C., ... & Porter, J. D. (2006). Effect of a structural intervention for the prevention of intimate-partner violence and HIV in rural South Africa: a cluster randomised trial. The lancet, 368(9551), 1973-1983.

Austrian, K., & Muthengi, E. (2014). Can economic assets increase girls' risk of sexual harassment? Evaluation results from a social, health and economic asset-building intervention for vulnerable adolescent girls in Uganda. Children and Youth Services Review, 47, 168-175.

Dunbar, M. S., Maternowska, M. C., Kang, M. S. J., Laver, S. M., Mudekunye-Mahaka, I., & Padian, N. S. (2010). Findings from SHAZ!: a feasibility study of a microcredit and life-skills HIV prevention intervention to reduce risk among adolescent female orphans in Zimbabwe. Journal of prevention & intervention in the community, 38(2), 147-161.

Sherman, S. G., German, D., Cheng, Y., Marks, M., & Bailey-Kloche, M. (2006). The evaluation of the JEWEL project: an innovative economic enhancement and HIV prevention intervention study targeting drug using women involved in prostitution. AIDS care, 18(1), 1-11.

7. PLOS authors have the option to publish the peer review history of their article (what does this mean?). If published, this will include your full peer review and any attached files.

Reviewer #1: No

Reviewer #3: No

Reviewer #4: No

---

## [Author Response · Author response to Decision Letter 1]

18 Mar 2023

Please see response letter attached.

---

## [Editor Report · Decision Letter 2]

4 Apr 2023

Bridges-Round 2: A study protocol to examine the longitudinal HIV risk prevention and care continuum outcomes among orphaned youth transitioning to young adulthood

PONE-D-22-11909R2

 Dear Author (s), 

We’re pleased to inform you that your manuscript has been judged scientifically suitable for publication and will be formally accepted for publication once it meets all outstanding technical requirements.

Kind regards,

Asrat Genet Amnie, MD, EdD, MPH, MBA

Academic Editor

PLOS ONE

---

## [Editor Report · Acceptance letter]

20 Apr 2023

PONE-D-22-11909R2 

Bridges-Round 2: A study protocol to examine the longitudinal HIV risk prevention and care continuum outcomes among orphaned youth transitioning to young adulthood. 

Dear Dr. Nabunya:

I'm pleased to inform you that your manuscript has been deemed suitable for publication in PLOS ONE. Congratulations! Your manuscript is now with our production department. 

Kind regards, 

on behalf of

Dr. Asrat Genet Amnie 

Academic Editor

PLOS ONE